# Reconstructing Parameters of Spreading Models from Partial Observations

**Andrey Y. Lokhov**
Center for Nonlinear Studies and Theoretical Division T-4
Los Alamos National Laboratory, Los Alamos, NM 87545, USA
lokhov@lanl.gov

## Abstract

Spreading processes are often modelled as a stochastic dynamics occurring on top of a given network with edge weights corresponding to the transmission probabilities. Knowledge of veracious transmission probabilities is essential for prediction, optimization, and control of diffusion dynamics. Unfortunately, in most cases the transmission rates are unknown and need to be reconstructed from the spreading data. Moreover, in realistic settings it is impossible to monitor the state of each node at every time, and thus the data is highly incomplete. We introduce an efficient dynamic message-passing algorithm, which is able to reconstruct parameters of the spreading model given only partial information on the activation times of nodes in the network. The method is generalizable to a large class of dynamic models, as well to the case of temporal graphs.

## 1  Introduction

Knowledge of the underlying parameters of the spreading model is crucial for understanding the global properties of the dynamics and for development of effective control strategies for an optimal dissemination or mitigation of diffusion [1, 2]. However, in many realistic settings effective transmission probabilities are not known a priori and need to be recovered from a limited number of realizations of the process. Examples of such situations include spreading of a disease [3], propagation of information and opinions in a social network [4], correlated infrastructure failures [5], or activation cascades in biological and neural networks [6]: precise model and parameters, as well as propagation paths are often unknown, and one is left at most with several observed diffusion traces. It can be argued that for many interesting systems, even the functional form of the dynamic model is uncertain. Nevertheless, the reconstruction problem still makes sense in this case: the common approach is to assume some simple and reasonable form of the dynamics, and recover the parameters of the model which explain the data in the most accurate and minimalistic way; this is crucial for understanding the basic mechanisms of the spreading process, as well as for making further predictions without overfitting. For example, if only a small number of samples is available, a few-parameter model should be used.

In practice, it is very costly or even impossible to record the state of each node at every time step of the dynamics: we might only have access to a subset of nodes, or monitor the state of the system at particular times. For instance, surveys may give some information on the health or awareness of certain individuals, but there is no way to get a detailed account for the whole population; neural avalanches are usually recorded in cortical slices, representing only a small part of the brain; it is costly to deploy measurement devices on each unit of a complex infrastructure system; finally, hidden nodes play an important role in the artificial learning architectures. This is precisely the setting that we address in this article: reconstruction of parameters of a propagation model in the presence of nodes with hidden information, and/or partial information in time. It is not surprising that this

challenging problem turns out to be notably harder then its detailed counterpart and requires new algorithms which would be robust with respect to missing observations.

**Related work.** The inverse problem of network and couplings reconstruction in the dynamic setting has attracted a considerable attention in the past several years. However, most of the existing works are focused on learning the propagation networks under the assumption of availability of full diffusion information. The papers [7, 8, 9, 10] developed inference methods based on the maximization of the likelihood of the observed cascades, leading to distributed and convex optimization algorithms in the case of continuous and discrete dynamics, principally for the variants of the independent cascade (IC) model [11]. These algorithms have been further improved under the sparse recovery framework [12, 13], particularly efficient for structure learning of treelike networks. A careful rigorous analysis of these likelihood-based and alternative [14, 15] reconstruction algorithms give an estimation of the number of observed cascades required for an exact network recovery with high probability. Precise conditions for the parameters recovery at a given accuracy are still lacking. The fact that the aforementioned algorithms rely on the fully observed spreading history represents an important limitation in the case of incomplete information. The case of missing time information has been addressed in two recent papers: focusing primarily on tree graphs, [16] studied the structure learning problem in which only initial and final spreading states are observed; [17] addressed the network reconstruction problem in the case of partial time snapshots of the network, using relaxation optimization techniques and assuming that full probabilistic trace for each node in the network is available. A standard technique for dealing with incomplete data involves maximizing the likelihood marginalized over the hidden information; for example, this approach has been used in [18] for identifying the diffusion source. In what follows, we use this method for benchmarking our results.

**Overview of results.** In this article, we propose a different algorithm, based on recently introduced dynamic message-passing (DMP) equations for cascading processes [19, 20], which will be referred to as DMPREC (DMP-reconstruction) throughout the text. Making use of all available information, it yields significantly more accurate reconstruction results, outperforming the likelihood method and having a substantially lower algorithmic complexity, independent on the number of nodes with unobserved information. More generally, the DMPREC framework can be easily adapted to allow reconstruction of heterogeneous transmission probabilities in a large class of cascading processes, including the IC and threshold models, SIR and other epidemiological models, rumor spreading dynamics, etc., as well as for the processes occurring on dynamically-changing networks.

## 2   Problem formulation

**Model.** For the sake of simplicity and definiteness, we assume that cascades follow the dynamics of stochastic susceptible-infected (SI) model in discrete time, defined on a network $G = (V, E)$ with set of nodes denoted by $V$ and set of directed edges $E$ [3]. Each node $i \in V$ at times $t = 1, 2, \ldots, T$ can be in either of two states: susceptible ($S$) or infected ($I$). At each time step, node $i$ in the $I$ state can activate one of its susceptible neighbors $j$ with probability $\alpha_{ij}$[1] . The dynamics is non-recurrent: once the node is activated (infected), it can never change its state back to susceptible. In what follows, the network $G$ is supposed to be known.

**Incomplete observations and inference problem.** We assume that the input is formed from $M$ independent cascades, where a cascade $\Sigma^c$ is defined as a collection of activation times of nodes in the network $\{\tau_i^c\}_{i \in V}$. Each cascade is observed up to the final observation time $T$. Notice that $T$ is an important parameter: intuitively, the larger is $T$, the more information is contained in cascades, and the less samples are needed. We assume that $T$ is given and fixed, being related to the availability of the finite-time observation window. If node $i$ in cascade $c$ does not get activated at a certain time prior to the horizon $T$, we put by definition $\tau_i^c = T$; hence, $\tau_i^c = T$ means that node $i$ changes its state at time $T$ or later. The full information on the cascades $\Sigma = \cup_c \Sigma^c$ is divided into the observed part, $\Sigma_{\mathcal{O}}$, and the hidden part $\Sigma_{\mathcal{H}}$. Thus, in general $\Sigma_{\mathcal{O}}$ contains only a subset of activation times in $\mathcal{T} \in [0, T]$ for a part of observed nodes in the network $\mathcal{O} \in V$. The task is to reconstruct the

couplings $\{\alpha_{ij}^*\}_{(ij)\in E} \equiv G_{\alpha^*}$, where $G_{\alpha^*}$ with a star denotes the original transmission probabilities that have been used to generate the data.

**Maximization of the likelihood.** Similarly to the formulations considered in [7, 8, 10], it is possible to explicitly write the expression for the likelihood of the discrete-time SI model in the case of fully available information $\Sigma_{\mathcal{O}} = \Sigma$ under the assumption that the data has been generated using the couplings $G_\alpha$:

$$P(\Sigma \mid G_\alpha) = \prod_{i\in V} \prod_{1\leq c\leq M} P_i(\tau_i^c \mid \Sigma^c, G_\alpha), \tag{1}$$

with

$$P_i(\tau_i^c \mid \Sigma^c, G_\alpha) = \left( \prod_{t'=0}^{\tau_i^c-2} \prod_{k\in\partial i} (1 - \alpha_{ki}\mathbb{1}_{\tau_k^c\leq t'}) \right) \left[ 1 - \left( \prod_{k\in\partial i} (1 - \alpha_{ki}\mathbb{1}_{\tau_k^c\leq \tau_i^c-1}) \right) \mathbb{1}_{\tau_i^c<T} \right], \tag{2}$$

where $\partial i$ denotes the set of neighbors of node $i$ in the graph $G$, and $\mathbb{1}$ is the indicator function. The expression (2) has the following meaning: the probability that node $i$ has been activated at time $\tau_i$ given the activation times of its neighbors is equal to the probability that the activation signal has not been transmitted by any infected neighbor of $i$ until the time $\tau_i - 2$ (first term in the product), and that at least one of the active neighbors actually transmitted the infection at time $\tau_i - 1$ (second term). A straightforward adaptation of the NETRATE algorithm, suggested in [8], to the present setting implies that the estimation of the transmission probabilities $\widehat{G}_{\alpha^*}$ is obtained as a solution of the convex optimization problem

$$\widehat{G}_{\alpha^*} = \arg\min\left(-\ln P(\Sigma \mid G_\alpha)\right), \tag{3}$$

which can be solved locally for each node $i$ and its neighborhood due to the factorization of the likelihood (1) under assumption of asymmetry of the couplings. In the case of partial observations, the optimization problem (3) is not well defined since it requires the full knowledge of activation times for each node. A simple and natural extension of this scheme, which we will refer to as the maximum likelihood estimator (MLE), is to consider the likelihood function marginalized over unknown activation times:

$$P(\Sigma_{\mathcal{O}} \mid G_\alpha) = \sum_{\{\tau_h^c\},h\in\mathcal{H}} P(\Sigma \mid G_\alpha). \tag{4}$$

An exact evaluation of (4) is a computationally hard high-dimensional integration problem with complexity proportional to $T^H$ in the presence of $H$ nodes with hidden information. In order to correct for this fact, we propose a heuristic scheme which we denote as the heuristic two-stage (HTS) algorithm. The idea of HTS consists of completing the missing part $\{\tau_h^c\}_{h\in\mathcal{H}}$ of the cascades at each step of the optimization process with the most probable values according to the current estimation of the couplings $\widehat{G}_\alpha$, $\widehat{\Sigma}_{\mathcal{H}} = \arg\max P(\Sigma \mid \widehat{G}_\alpha)$, and solving the optimization problem (3) using the full information on the cascades $\Sigma = \Sigma_{\mathcal{O}} \cup \widehat{\Sigma}_{\mathcal{H}}$; these two alternating steps are iterated until the global convergence of the algorithm. An exact (brute-force) estimation of $\widehat{\Sigma}_{\mathcal{H}}$ requires an exponential number of operations $T^H$, as the original MLE formulation. However, we found that in practice the computational time can be significantly reduced with the use of the Monte Carlo sampling. The corresponding approximation is based on the observation that the likelihood (1) is non-zero only for $\{\tau_i^c\}_{i\in V}$ forming possible (realizable) cascades. Hence, for each $c$, we sample $L_{H,T}$ auxiliary cascades, and choose the set of $\{\tau_h^c\}_{h\in\mathcal{H}}$ maximizing (1). $L_{H,T}$ is typically a large sampling parameter, growing with $T$ and $H$ to ensure a proper convergence. This procedure leads to an algorithm with a complexity $O(NM|E|^2 L_{H,T})$ at each step of the optimization, where $|E|$ denotes the number of edges; see the journal version of the paper [21] for a more in-depth discussion.

Hence, both MLE and HTS algorithms are practically intractable; the remaining part of the paper is devoted to the development of an accurate algorithm with a polynomial-time computational complexity for this hard problem. The next section introduces dynamic message-passing equations which serve as a basis for such algorithm.

## 3 Dynamic message-passing equations.

The dynamic message-passing equations for the SI model in continuous [19] and discrete [20] settings allow to compute marginal probabilities that node $i$ is in the state $S$ at time $t$:

$$P_S^i(t) = P_S^i(0) \prod_{k\in\partial i} \theta^{k\to i}(t) \tag{5}$$

for $t > 0$ and a given initial condition $P_S^i(0)$. The variables $\theta^{k \to i}(t)$ represent the probability that node $k$ did not pass the activation signal to the node $i$ until time $t$. The intuition behind the key Equation (5) is that the probability of node $i$ to be susceptible at time $t$ is equal to the probability of being in the $S$ state at initial time times the probability that neither of its neighbors infected it until time $t$. The quantities $\theta^{k \to i}(t)$ can be computed iteratively using the following expressions:

$$\theta^{k \to i}(t) = \theta^{k \to i}(t-1) - \alpha_{ki}\phi^{k \to i}(t-1), \tag{6}$$

$$\phi^{k \to i}(t) = (1 - \alpha_{ki})\phi^{k \to i}(t-1) + P_S^k(0)\left( \prod_{l \in \partial k \backslash i} \theta^{l \to k}(t-1) - \prod_{l \in \partial k \backslash i} \theta^{l \to k}(t) \right), \tag{7}$$

where $\partial k \backslash i$ denotes the set of neighbors of $k$ excluding $i$. The Equation (6) translates the fact that $\theta^{k \to i}(t)$ can only decrease if the infection is actually transmitted along the directed link $(ki) \in E$; this happens with probability $\alpha_{ki}$ times $\phi^{k \to i}(t-1)$ which denotes the probability that node $k$ is in the state $I$ at time $t$, but has not transmitted the infection to node $i$ until time $t-1$. The Equation (7), which allows to close the system of dynamic equations, describes the evolution of probability $\phi^{k \to i}(t)$: at time $t-1$, it decreases if the infection is transmitted (first term in the sum), and increases if node $k$ goes from the state $S$ to the state $I$ (difference of terms 2 and 3). Note that node $i$ is excluded from the corresponding products over $\theta$-variables because this equation is conditioned on the fact that $i$ is in the state $S$, and therefore can not infect $k$. The Equations (6) and (7) are iterated in time starting from initial conditions $\theta^{i \to j}(0) = 1$ and $\phi^{i \to j}(0) = 1 - P_S^i(0)$ which are consistent with the definitions above. The name "DMP equations" comes from the fact the whole scheme can be interpreted as the procedure of passing "messages" along the edges of the network.

**Theorem 1.** *DMP equations for the SI model, defined by Equations* (5)-(7)*, yield exact marginal probabilities on tree networks. On general networks, the quantities $P_S^i(t)$ give lower bound on values of marginal probabilities.*

*Proof Sketch.* The exactness of solution on tree graphs immediately follows from the fact that the DMP equations can be derived from belief propagation equations on time trajectories [20], which provide exact marginals on trees. The fact that $P_S^i(t)$ computed according to (5) represent a lower bound on marginal probabilities in general networks can be derived from a counting argument, considering multiple infection paths on a loopy graph which contribute to the computation of $P_S^i(t)$, effectively lowering its value through the Equation (5); the proof technique is borrowed from [19], where similar dynamic equations in the continuous-time case have been considered. $\square$

Using the definition (5) of $P_S^i(t)$, it is convenient to define the marginal probability $m^i(t)$ of activation of node $i$ at time $t$:

$$m^i(t) = P_S^i(0)\left[ \prod_{k \in \partial i} \theta^{k \to i}(t-1) - \prod_{k \in \partial i} \theta^{k \to i}(t) \right]. \tag{8}$$

As it often happens with message-passing algorithms, although being exact only on tree networks, DMP equations provide accurate results even on loopy networks. An example is provided in the Figure 1, where the DMP-predicted marginals are compared with the values obtained from extensive simulations of the dynamics on a network of retweets with $N = 96$ nodes [22]. This observation will allow us to use DMP equations as a suitable approximation tool on general networks. In the next section we describe an efficient reconstruction algorithm, DMPREC, which is based on the resolution of the dynamics given by DMP equations and makes use of all available information.

## 4    Proposed algorithm: DMPREC

**Probability of cascades and free energy.** The marginalization over hidden nodes in (4) creates a complex relation between couplings in the whole graph, resulting in a non-explicit expression. The main idea behind the DMPREC algorithm is to approximate the likelihood of observed cascades (4) through the marginal probability distributions (5) and (8):

$$P(\Sigma_\mathcal{O} \mid G_\alpha) \approx \prod_{c=1}^{M} \prod_{i \in \mathcal{O}} \left[ m^i(\tau_i^c \mid G_\alpha)\mathbb{1}_{\tau_i^c \leq T} + P_S^i(\tau_i^c \mid G_\alpha)\mathbb{1}_{\tau_i^c = T} \right]. \tag{9}$$

The expression (9) is at the core of the suggested algorithm. As there is no tractable way to compute exactly the joint probability of partial observations, we approximate it using a mean-field-type

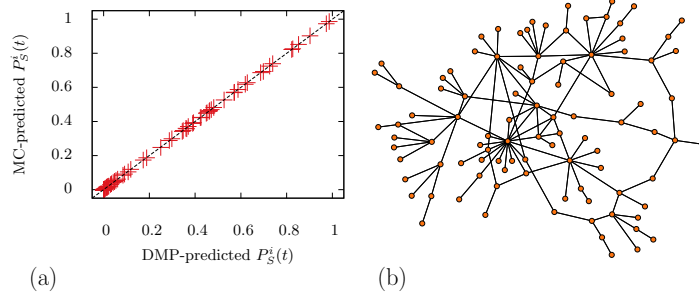

(a)                                            (b)

Figure 1: Illustration of the accuracy of DMP equations on a network of retweets with $N = 96$ nodes [22]. (a) Comparison of DMP-predicted $P_S^i(t)$ with $P_S^i(t)$ estimated from $10^6$ runs of Monte Carlo simulations with $t = 10$ and one infected node at initial time. The couplings $\{\alpha_{ij}\}$ have been generated uniformly at random in the range $[0, 1]$. (b) Visualization of the network topology created with the Gephi software.

approach as a product of marginal probabilities provided by the dynamic message-passing equations. The reasoning behind this approach is that each marginal is expressed through an average of all possible realizations of dynamics with a given initial condition; this is in contrast with the likelihood function which considers only particular instance realized in the given cascade. Therefore, equation (9) summarizes the effect of different propagation paths, and the maximization of this probability function will yield the most likely consensus between the ensemble of couplings in the network. Precisely this key property makes the reconstruction possible in the case involving nodes with hidden information via maximization of the objective (9) which can be interpreted as a cost function representing the product of individual probabilities of activation taken precisely at the value of the observed infection times. Starting from this expression, one can define the associated "free energy":

$$f_{\text{DMP}} = -\ln P(\Sigma_\mathcal{O} \mid G_\alpha) = \sum_{i \in \mathcal{O}} f_{\text{DMP}}^i, \tag{10}$$

where $f_{\text{DMP}}^i = -\sum_c \ln\left[m^i(\tau_i^c)\mathbb{1}_{\tau_i^c \leq T-1} + P_S^i(T-1)\mathbb{1}_{\tau_i^c = T}\right]$. In the last expression for $f_{\text{DMP}}^i$ we used the fact that $m^i(T) + P_S^i(T) = P_S^i(T-1)$. Our goal is to minimize the free energy (10) with respect to $\{\alpha_{ij}\}_{(ij)\in E}$. A similar approach has been previously outlined by [23] as a way to learn homogeneous couplings in the spreading source inference algorithm. In order to carry out this optimization task, we need to develop an efficient way of gradient evaluation.

**Computation of the gradient.** The gradient of the free energy reads (note that the indicator functions point to disjoint events):

$$\frac{\partial f_{\text{DMP}}^i}{\partial \alpha_{rs}} = -\sum_c \left[\frac{\partial m^i(\tau_i^c \mid G_\alpha)/\partial \alpha_{rs}}{m^i(\tau_i^c \mid G_\alpha)}\mathbb{1}_{\tau_i^c \leq T-1} + \frac{\partial P_S^i(T-1 \mid G_\alpha)/\partial \alpha_{rs}}{P_S^i(T-1 \mid G_\alpha)}\mathbb{1}_{\tau_i^c = T}\right], \tag{11}$$

where the derivatives of the marginal probabilities can be computed explicitly by taking the derivative of the DMP equations (5)-(8). Let us denote $\partial \theta^{k\to i}(t)/\partial \alpha_{rs} \equiv p_{rs}^{k\to i}(t)$ and $\partial \phi^{k\to i}(t)/\partial \alpha_{rs} \equiv q_{rs}^{k\to i}(t)$. Since the dynamic messages at initial time $\{\theta^{i\to j}(0)\}$ and $\{\phi^{i\to j}(0)\}$ are independent on the couplings, we have $p_{rs}^{k\to i}(0) = q_{rs}^{k\to i}(0) = 0$ for all $k$, $i$, $r$, $s$, and these quantities can be computed iteratively using the analogues of the Equations (6) and (7):

$$p_{rs}^{k\to i}(t) = p_{rs}^{k\to i}(t-1) - \alpha_{ki}q_{rs}^{k\to i}(t-1) - \phi^{k\to i}(t-1)\mathbb{1}_{k=r,i=s}, \tag{12}$$

$$q_{rs}^{k\to i}(t) = (1-\alpha_{ki})q_{rs}^{k\to i}(t-1) - \phi^{k\to i}(t-1)\mathbb{1}_{k=r,i=s}$$
$$+ P_S^k(0)\sum_{l\in\partial k\backslash i} p_{rs}^{l\to k}(t-1)\prod_{n\in\partial k\backslash\{i,l\}}\theta^{n\to k}(t-1) - P_S^k(0)\sum_{l\in k\backslash i} p_{rs}^{l\to k}(t)\prod_{n\in\partial k\backslash\{i,l\}}\theta^{n\to k}(t). \tag{13}$$

Using these quantities, the derivatives of the marginals entering in Equation (11) can be written as

$$\frac{\partial P_S^i(t)}{\partial \alpha_{rs}} = P_S^i(0)\sum_{k\in\partial i} p_{rs}^{k\to i}(t)\prod_{l\in\partial i\backslash k}\theta^{l\to i}(t), \qquad \frac{\partial m^i(t)}{\partial \alpha_{rs}} = \frac{\partial P_S^i(t-1)}{\partial \alpha_{rs}} - \frac{\partial P_S^i(t)}{\partial \alpha_{rs}}. \tag{14}$$

The following observation shows that at least on tree networks, corresponding to the regime in which DMP equations have been derived, the values of the original transmission probabilities $G_{\alpha^*}$ correspond to the point in which the gradient of the free energy takes zero value.

**Claim 1.** *On a tree network, in the limit of large number of samples $M \to \infty$, the derivative of the free energy is equal to zero at the values of couplings $G_{\alpha^*}$ used for generating cascades.*

*Proof.* Let us first look at samples originating from the same initial condition. According to Theorem 1, the DMP equations are exact on tree graphs, and hence it is easy to see that

$$\lim_{M \to \infty} f^i_{\text{DMP}} = -\sum_{t \leq T-1} m^i(t \mid G_{\alpha^*}) \ln m^i(t \mid G_\alpha) - P^i_S(T-1 \mid G_{\alpha^*}) \ln P^i_S(T-1 \mid G_\alpha). \quad (15)$$

Therefore,

$$\lim_{M \to \infty} \frac{\partial f^i_{\text{DMP}}}{\partial \alpha_{rs}} \big|_{G_{\alpha^*}} = -\frac{\partial}{\partial \alpha_{rs}} \left[ \sum_{t \leq T-1} m^i(t \mid G_{\alpha^*}) + P^i_S(T-1 \mid G_{\alpha^*}) \right] = 0,$$

since the expression inside the brackets sums exactly to one. This result trivially holds by summing up samples with different initial conditions. Combination of this result with the definition (10) completes the proof. $\qquad \square$

The DMPREC algorithm consists of running the message-passing equations for the derivatives of the dynamic variables (12), (13) in parallel with DMP equations (5)-(7), allowing for the computation of the gradient of the free energy (11) through (14), which is used afterwards in the optimization procedure. Let us analyse the computational complexity of each step of parameters update. The number of runs is equal to the number of distinct initial conditions in the ensemble of observed cascades, so if all $M$ cascades start with distinct initial conditions, the complexity of the DMPREC algorithm is equal to $O(|E|^2 T M)$ for each step of the update of $\{\alpha_{rs}\}_{(rs)\in E}$. Hence, in a typical situation where each cascade is initiated at one particular node, the number of runs will be limited by $N$, and the overall update-step complexity of DMPREC will be $O(|E|^2 T N)$.

**Missing information in time.** On top of inaccessible nodes, the state of the network can be monitored at a lower frequency compared to the natural time scale of the dynamics. It is easy to adapt the algorithm to the case of observations at $K$ time steps $\mathcal{T} \equiv \{t^k\}_{k\in[1,K]}$. Since the activation time $\tau^c_i$ of node $i$ in cascade $c$ is now known only up to the interval $[t^{k^c_i} + 1, t^{k^c_i+1}] \equiv \delta_{k^c_i}$, where $t^{k^c_i} < \tau^c_i \leq t^{k^c_i+1}$, one should maximize $\sum_{t \in \delta_{k^c_i}} m^i(t) = P^i_S(t^{k^c_i}) - P^i_S(t^{k^c_i+1}) \equiv \Delta_{k^c_i} P^i_S(t \mid G_\alpha)$ instead of $m^i(\tau^c_i)$ in this case. This leads to obvious modifications to the expressions (10) and (11), using the differences of derivatives at corresponding times instead of one-step differences as in (14). For instance, if the final time is not included in the observations, we have

$$f^i_{\text{DMP}} = -\sum_c \ln \left[ \Delta_{k^c_i} P^i_S(t \mid G_\alpha) \right], \quad \frac{\partial f^i_{\text{DMP}}}{\partial \alpha_{rs}} = -\sum_c \left[ \frac{\partial \Delta_{k^c_i} P^i_S(t \mid G_\alpha)/\partial \alpha_{rs}}{\Delta_{k^c_i} P^i_S(t \mid G_\alpha)} \right].$$

# 5 Numerical results

We evaluate the performance of the DMPREC algorithm on synthetic and real-world networks under assumption of partial observations. In numerical experiments, we focus primarily on the presence of inaccessible nodes, which is a more computationally difficult case compared to the setting of missing information in time. An example involving partial time observations is shown in section 5.1.

## 5.1 Tests with synthetic data

**Experimental setup.** In the tests described in this section, the couplings $\{\alpha_{ij}\}$ are sampled uniformly in the range $[0, 1]$, the final observation time is set to $T = 10$. Each cascade is generated using a discrete-time SI model defined in section 2 from randomly selected sources. In the case of inaccessible nodes, the activation times data is hidden in all the samples for $H$ randomly selected nodes. We use the likelihood methods for benchmarking the accuracy of our approach. The MLE algorithm introduced above is not tractable even on small graphs, therefore we compare the results of DMPREC with

the HTS algorithm outlined in the section 2. Still, HTS has a very high computational complexity, and therefore we are bound to run comparative tests on small graphs: a connected component of an artificially-generated network with $N = 20$, sampled using a power-law degree distribution, and a real directed network of relationships in a New England monastery with $N = 18$ nodes [24]. Both algorithms are initialized with $\alpha_{ij} = 0.5$ for all $(ij) \in E$. The accuracy of reconstruction is assessed using the $\ell_1$ norm of the difference between reconstructed and original couplings, normalized over the number of directed edges in the graph[2]. Intuitively, this measure gives an average expected error for each parameter $\alpha_{ij}$.

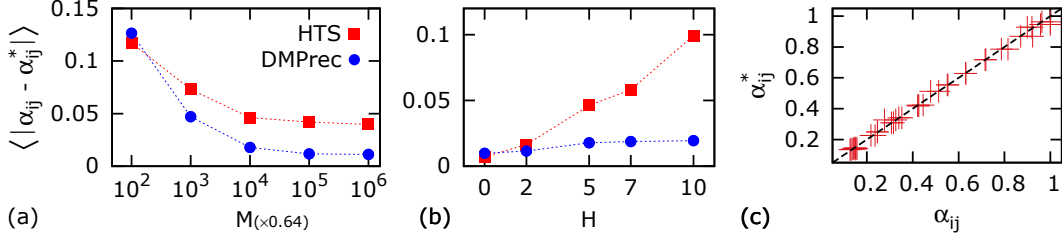

(a)   (b)   (c)

Figure 2: Tests for DMPREC and HTS on a small power-law network: (a) for fixed number of nodes with unobserved information $H = 5$, (b) for fixed number of samples $M = 6400$. (c) Scatter plot of $\{\alpha_{ij}\}$ obtained with DMPREC versus original parameters $\{\alpha_{ij}^*\}$ in the case of missing information in time with $M = 6400$, $T = 10$; the state of the network is observed every other time step.

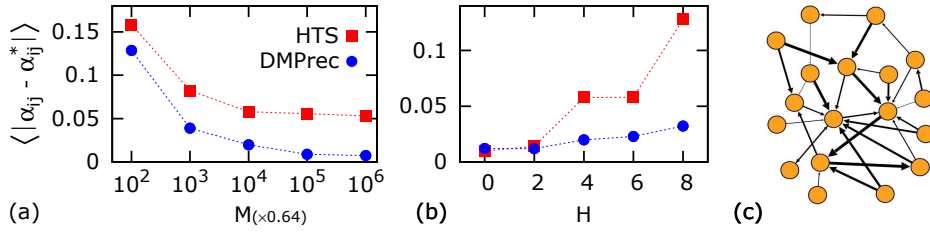

(a)   (b)   (c)

Figure 3: Numerical results for the real-world Monastery network of [24]: (a) for fixed number of nodes with unobserved information $H = 4$, (b) for fixed number of samples $M = 6400$. (c) The topology of the network (thickness of edges proportional to $\{\alpha_{ij}^*\}$ used for generating cascades).

**Results.** In the Figure 2 we present results for a small power-law network with short loops, which is not a favorable situation for DMP equations derived in the treelike approximation of the graph. Figures 2 (a) and 2 (b) show the dependence of an average reconstruction error as a function of $M$ (for fixed $H/N = 0.25$) and $H$ (for fixed $M = 6400$), respectively. DMPREC clearly outperforms the HTS algorithm, yielding surprisingly accurate reconstruction of transmission probabilities even in the case where a half of network nodes do not report any information. Most importantly, DMPREC achieves reconstruction with a significantly lower computational time: for example, while it took more than 24 hours to compute the point corresponding to $H = 4$ and $M = 6400$ with HTS (MLE at this test point took several weeks to converge), the computation involving DMPREC converged to the presented level of accuracy in less than 10 minutes on a standard laptop. These times illustrate the hardness of the learning problem involving incomplete information.

We have also used this case study network to test the estimation of transmission probabilities with the DMPREC algorithm when the state of the network is recorded only at a subset of times $\mathcal{T} \in [0, T]$. Results for the case where every other time stamp is missing are given in the Figure 2 (c): couplings estimated with DMPREC are compared to the original values $\{\alpha_{ij}^*\}$; despite the fact that only 50% of time stamps are available, the inferred couplings show an excellent agreement with the ground truth.

Equivalent results for the real-world relationship network extracted from the study [24] and containing both directed and undirected links, are shown in the Figure 3; an ability of DMPREC to capture the mutual dependencies of different couplings through dynamic correlations is even more pronounced in this case, with almost perfect reconstruction of couplings for large $M$ and a rather weak dependence

on the number of nodes with removed observations. We have run tests on larger synthetic networks which show similar reconstruction results for DMPREC, but where comparisons with the likelihood method could not be carried out. In the next section we focus on an application involving real-world data which represents a more interesting and important case for the validation of the algorithm.

## 5.2   Test with a real-world data

As a proxy for the real statistics, we used the data provided by the Bureau of Transportation Statistics [25], from which we reconstructed a part of the U.S. air transportation network, where airports are the nodes, and directed links correspond to traffic between them. The reason behind this choice is based on the fact that the majority of large-scale influenza pandemics over the past several decades represented the air-traffic mediated epidemics. For illustration purposes, we selected top $N = 30$ airports ranked according to the total number of passenger enplanements and commonly classified as large hubs, and extracted a sub-network of flights between them. The weight of each edge is defined by the annual number of transported passengers, aggregated over multiple routes; we have pruned links with a relatively low traffic – below $10\%$ of the traffic level on the busiest routes, so that the total number of remaining directed links is $|E| = 210$. The final weights are based on the assumption that the probability of infection transmission is proportional to the flux; the weights have been renormalized accordingly so that the busiest route received the coupling $\alpha_{ij} = 0.5$. The resulting network is depicted in the Figure 4 . We have generated $M = 10,000$ independent cascades in this network, and have hidden the information at $H = 15$ nodes (50% of airports) selected at random. We observe that even with a significantly large portion of missing information, the reconstructed parameters show a good agreement with the original ones.

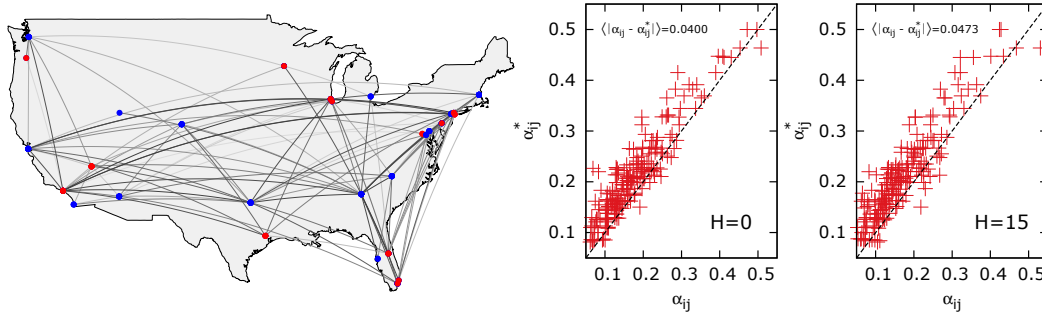

Figure 4: Left: Sub-network of flights between major U.S. hubs, where the thickness of edges is proportional to the aggregated traffic between them; nodes which do not report information are indicated in red. Right: Scatter plots of reconstructed $\{\alpha_{ij}\}$ versus original $\{\alpha_{ij}^*\}$ couplings for $H = 0$ and $H = 15$ and $M = 10,000$.

## 6   Conclusions and path forward

From the algorithmic point of view, inference of spreading parameters in the presence of nodes with incomplete information considerably complicates the problem because the reconstruction can no longer be performed independently for each neighborhood. In this paper, it is shown how the dynamic interdependence of parameters can be exploited in order to be able to recover the couplings in the setting involving hidden information. Let us discuss several directions for future work. DMPREC can be straightforwardly generalized to more complicated spreading models using a generic form of DMP equations [20] and the key approximation ingredient (9), as well as adapted to the case of temporal graphs by encoding network dynamics via time-dependent coefficients $\alpha_{ij}(t)$, which might be more appropriate in certain real situations. It would also be useful to extend the present framework to the case of continuous dynamics using the continuous-time version of DMP equations of [19]. An important direction would be to generalize the learning problem beyond the assumption of a known network, and formulate precise conditions for detection of hidden nodes and for a perfect network recovery in this case. Finally, in the spirit of active learning, we anticipate that DMPREC could be helpful for the problems involving an optimal placement of observes in the situations where collection of full measurements is costly.

**Acknowledgements.** The author is grateful to M. Chertkov and T. Misiakiewicz for discussions and comments, and acknowledges support from the LDRD Program at Los Alamos National Laboratory by the National Nuclear Security Administration of the U.S. Department of Energy under Contract No. DE-AC52-06NA25396.

## Footnotes

[1]We chose this two-state model since it has slightly more general dynamic rules compared to the popular IC model [11] with an additional restriction: a node infected at time $t$ has a single chance to activate its susceptible neighbors at time step $t+1$, while further infection attempts in subsequent rounds are not allowed. The DMPREC method presented below can be easily applied to the case of IC model by noticing that it corresponds to the SIR model with a recovery probability equal to one, for which the DMP equations are known [20].

[2]Note that this measure excludes those few parameters which are impossible to reconstruct: e.g. no algorithm can learn the coupling associated with the ingoing edge of the hidden node located at the leaf of a network.

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
