[Reviews · NeurIPS 2016]

Reviewer 1

Summary

The paper describes a method for estimating transmission probabilities for diffusion over a network, when only partial information is available. In contrast with full information scenario (where each node's neighborhood can be processed independently), here we need to learn the parameters for the unobserved nodes using information from other nodes. The authors propose an estimation algorithm based on message passing and show that it is better in accuracy and computational efficiency than MLE baseline.

Qualitative Assessment

The paper tackles a general problem of parameter estimation in communication graphs that may apply to different domains with network diffusion. The algorithm is reasonable and follows from likelihood and mean-field approaches. The gains in computational efficiency compared to MLE baseline are impressive (authors do not cite or compare to any other method, so I am assuming that's the most reasonable baseline). I understand that MLE approaches are slow and therefore testing on larger networks can be impractical. However, the paper evaluates their algorithm on small networks (N=30). Therefor, it is not clear how useful this algorithm will be in practice, where a sizeable number of nodes may be missing or when the missing nodes may not be at random wrt. the structure of the network. Here are some experiments that will be useful to see: 1. Instead of removing nodes uniformly at random, the nodes may be removed in inverse proportion to their importance. This is a fairly likely scenario: we expect to have data for busy airports, but less so for smaller airports. 2. It might also be useful to understand how the underestimation varies as the "loopiness" of the graph increases, because it is realistic to expect graphs with loops. This could also be done by transforming edges in the available small-N networks. 3. How do the run time of the proposed algorithm scale with the size of the network in practice? In other words, what size of networks can this approach be applied empirically? Overall, the paper is well-written and experiments are clearly described. One minor detail: I'm assuming that the authors are using independent cascades as their diffusion model, but it will be good to specify it in section 5.1.

Confidence in this Review

2-Confident (read it all; understood it all reasonably well)


Reviewer 2

Summary

The goal of this paper is to infer the parameters of a susceptible-infected (SI) model on a known graph, in the face of partial information about the status of nodes over time. Since summing over all possible unobserved components of the network to get the actual likelihood is intractable, the MS. proposes approximating this by message-passing, borrowing from work on message-passing techniques for SI models. My sense from the presentation in the MS. is that the material in section 3 is old work, while the new stuff comes in section 4, especially in the efficient calculation of the gradient. The problem considered is interesting but extremely stylized: not only is SI rather a caricatue of any real epidemiological process, but it's assumed that there are many completely independent realizations of the epidemic process to average over. (In the "real data" example of section 5.2, results are reported using _ten thousand_ independent cascades over a fragment of the US airport network. But if each cascade is taken to represent, say, a flu season, the idea of having even _thirty_ independent cascades under similar conditions strains credulity.) Moreover, it is not at all clear that the present approach really would suggest a way forward to more realistic epidemic models and/or problem situations. Still, as an incremental contribution to this literature with reasonable numerics, it looks promising.

Qualitative Assessment

This seems like a reasonable contribution to advancing the state-of-the-art on a stylized problem; whether it could be a step to something more significant is less clear to me.

Confidence in this Review

2-Confident (read it all; understood it all reasonably well)


Reviewer 3

Summary

The paper deals with the problem of diffusion graph weighting, from activity data observed from the studied networks. More specifically, it focus on an adaptation of NetRate (which corresponds to an extension of the Independent Cascade model with delays of infection, here in the discrete case, following a geometric law on these transmission delays) to the case where only the activity of a subset of the nodes of the network can be monitored. The claim is that non observed nodes shouldn't be ignored when learning the transmission probabilities of the diffusion graph.

Qualitative Assessment

Very interesting and well written paper, which deals with an important problem when learning diffusion models: it may be impossible to observe the activity of every node of the network. For instance, if the source is Twitter, the streaming API only allows one to monitor the activity of about 4000 users over several millions. The proposed approach is elegant and well sounded. I enjoyed to review this paper that brought me new knowledge about learning in such complex situations. The proposal is maybe not too much innovative since it mostly employs already published techniques, but from my point of view the followed methodology deserves to be published in some main machine learning venue, at least for its pedagogical aspect. Moreover, a good theoretical analysis of the algorithm is given for the case of tree network structures. I have two concerns about the applicability of the method in real-world scenarios: - The model is based (as NetRate and a lot of other models) on the hypothesis that the earlier a node is infected the more likely it is to be. This is induced by the fact that a single parameter controls the activation probability and the time of this activation. In such models, the non-infected nodes are usually considered as infected at a horizon time T. My experience is that this parameter $T$ has a very great impact on the results and is very difficult to tune. However, considering asynchronous diffusion models such as Saito's ones leads to greatly more complex likelihood formulations. I do not know if the present work could be transposed in such a setting. - The model considers a geometric law for the time delays of transmission between nodes. This is a strong assumption about the regularities on transmission dynamics that may not hold in real life. Three questions/coments about the experiments: - What is the MonteCarlo method used for HTS ? From my point of view this should be an MCMC, such as Gibbs Sampling, since it has to consider activation times of observed infected nodes. - How are generated the cascades from the real world graph? Using a IC model with a geometric law on diffusion delays as considered in the paper ? I regret that real "real world" data have not been used here, as this does not allow to assess the model for real life tasks... - Only very small networks have been considered here. It would be nice to experiment it on greatly larger networks to assess how it scales For me the paragraph "missing information in time" should be removed since it presents very few interest: it appears as obvious that the algorithm can be easily adapted with a different discretization of the time. At least, it should be reworked (I spent more time to understand this small paragraph than the whole rest of the paper!): - what is the difference between $\cal K$ and k here ? For me, $k$ should be used in place of $\cal K$ everywhere - for me $\delta_k$ should be noted $\delta_i^c$, it is the interval in which $\tau_i^c$ is included. Then, it would allow one to skip the indicator function in the following equations, by giving $\Delta_{k_i^c}$ that would be more clear - line 215: the upperbound of $\tau_i^c$ should be $\tau^{k+1}$ rather than $\tau^{k}$ - Last but not least: I do not understand why non-infections are ignored here. It is claimed that, provided the final time T is not included in the observations, the second term of (9) can be ignored. I disagree with this: every non infected node at the end of the last observed step should be considered as not infected. Thus, a term $P_S^i$ should be given for them. Else, every probability is likely to be greatly over-estimated.. No ? Minor comments: - "Using expression (11)" line 199 is somewhat misleading (at least it confused me a little for the comprehension of the proof). From my point of view, it would be more clear without this indication. - Formula (9): \leq T-1 rather than \leq T in the first term, no ? (or strict inequality) - Line 75: "the its" - Formula 4 : I have some problems with the notation of the sum. For me the subscript notation for "each possible set of hidden activation times" is not correct here. - I would add a $\cal O$ as a subscript of \Sigma in formula (9) - Line 282: overestimated => underestimated (and if I am right the opposite operation line 280)

Confidence in this Review

3-Expert (read the paper in detail, know the area, quite certain of my opinion)


Reviewer 4

Summary

This paper considers the problem of reconstructing parameters of spreading models from partial observations. Because calculating the exact joint probability of partial observations is intractable, it uses the mean-field-type approximations along with the dynamic message passing to get the estimations of the parameters of a propagation model. Compared with the heuristic two-stage (HTS) algorithm, the proposed algorithm has more accurate reconstruction results, lower computational costs, and more robustness over the number of hidden nodes over synthetic data. It also shows meaningful results on real world air transportation data.

Qualitative Assessment

My main concern of this paper is that it lacks the comparison over the other approximation approaches. I totally understand the exact calculation is intractable for high dimensional data with partial observations. However, there are several choices of approximation methods. The author needs to justify or compare the chosen mean-field approximation over the other approaches such as expectation propagation approximation and sampling methods. I noticed sampling is used in the HST, but I want to know the comparison over a whole sampling design. In addition, in the title and introduction, the authors mentioned the proposed algorithm can apply to the spreading models. However, in the problem formulation section, they narrow down the scope to the stochastic susceptible-infected (SI) model. Does SI model represent all the other spreading models? What if the dynamics is recurrent? What about susceptible-infected-susceptible (SIS) model? I want to know if the proposed algorithm generalizes well with the other models. Also, in the real world influenza pandemics example in section 5.2, I think SIS model is more appropriate because people can recover from the influenza. Also, there are some missing details in the experiment part. For experimental setup in section 5.1, in the synthetic data, how many different graph structures are tested? How sparse are these graphs? In figure 3(b), why HTS behaves very differently when H grows from 4 to 6? Also, in the missing time step experiments, the “every other time stamp is missing” setup is too arbitrary and it is not conclusive. In figure 4, why the parameters are constantly overestimated? Can you explain it? At the end of the paper, there is a fake URL that intends to show the data set and the code. Did you include these materials in the Supplementary Files? I did not find them. If not, I think the fake URL is not necessary here. Detailed Comments: I think the description that the proposed algorithm is “independent on the number of nodes with unobserved information” (line 65) is not accurate as shown in Figure 2(b). I think they are not independent. In my personal opinion, the proof of theorem 1 is not necessary because I did not see new insights in this proof. It just repeats the existing proof. I did not find the definition of N in line 115. In line 136, it should be “the” instead of “t he”.

Confidence in this Review

2-Confident (read it all; understood it all reasonably well)


Reviewer 5

Summary

The paper provides a version of the message-passing algorithm in order to compute marginal probabilities that appear in the likelihood function marginalized over unknown activation times in the spreading process where the transmition rates are unknown parameters and a part of the cascade is hidden. The paper also provides numerical experiments both on synthetic data and on real-world data in order to show that the proposed method works better than the methods where the marginal (conditional) likelihood is evaluated exactly.

Qualitative Assessment

Although it is certainly a useful practice to apply message-passing algorithm to compute conditional likelihoods in the proposed setting of the paper, I think the paper lacks many important factors required for publication as I outline below: The paper is very weak in expressing the basic statistical concepts used in the paper: for example only on page 3: there is no need to adapt any algorithm to see that the MLE for G_alpha is the solution to Equation (3); you cannot refer to any scheme as maximum likelihood estimator (MLE) -- MLE has a very clear meaning, and although it is true that you are finding the MLE for the conditional likelihood (4), the provided explanation is extremely confusing in the least; you should be clear what Monte Carlo method you are using for estimation of Sigma_H (I assume some type of MCMCMLE?); etc. Although the paper claims to propose a new algorithm, it seems to me that all it does is to compute the conditional likelihood through computing the necessary marginal likelihoods by message-passing. So this seems not to be a new algorithm, but just a way to compute (relatively trivial) steps of the original algorithm faster. I do not see how Equations (6) and (7) are related to Theorem 1. I think you need another result explaining why (6) and (7) compute the expression in (5). The quality of English writing is very low: for example, "can be argued" not "an argued"; "consists of" not "consists in"; "denoted by" not "denoted as"; "mean ..." not "have a meaning ..."; "respectively" not "correspondingly"; etc.

Confidence in this Review

2-Confident (read it all; understood it all reasonably well)


Reviewer 6

Summary

The paper introduces a dynamic message-passing algorithm for spreading parameter reconstruction in a network from incomplete spreading data. The network structure is assumed to be known and the cascades follow the dynamics of the SI-model in discrete time. The model is evaluated on both, a synthetic- and a real-world data set. On the synthetic data set the algorithm is compared to a related algorithm (HTS). Its applicability to real-world problems is demonstrated by an experiment on the Bureau of Transportation Statistics data set.

Qualitative Assessment

The scope and goals of this work are clearly stated, the overall presentation is accessible. While the approach seems promising, the assumptions of knowing the entire network and the SI-model for the spreading dynamics seem (to me as a non-expert in this field) as strong limitations. While the experiments on the synthetic data seem rather weak (only a comparison to one other algorithm on a small data set), I did find the experiment on the real-world data interesting. I look forward of being corrected in the rebuttal period.

Confidence in this Review

1-Less confident (might not have understood significant parts)